# Reconstructive Social Innovation Cycles in Women-Led Initiatives in Rural Areas

Simo Sarkki [1,*] , Cristina Dalla Torre [2,3] , Jasmiini Fransala [1] , Ivana Živojinović [4,5] , Alice Ludvig [4,5] , Elena Górriz-Mifsud [6,7] , Mariana Melnykovych [6,8] , Patricia R. Sfeir [9] , Labidi Arbia [10] , Mohammed Bengoumi [10] , Houda Chorti [10] , Verena Gramm [2] , Lucía López Marco [11] , Elisa Ravazzoli [2] and Maria Nijnik [12]

1   Cultural Anthropology, University of Oulu, 90570 Oulu, Finland; Jasmiini.Fransala@oulu.fi
2   Institute for Regional Development, Eurac Research, 39100 Bolzano, Italy; cristina.dallatorre@eurac.edu (C.D.T.); verena.g.89@hotmail.com (V.G.); elisa.ravazzoli@eurac.edu (E.R.)
3   Department of Land, Environment, Agriculture and Forestry (TESAF), University of Padova, 35020 Legnaro, Italy
4   Institute of Forest, Environment and Natural Resource Policy, University of Natural Resources and Life Sciences (BOKU), 1180 Vienna, Austria; ivana.zivojinovic@boku.ac.at (I.Ž.); alice.ludvig@boku.ac.at (A.L.)
5   European Forest Institute, Forest Policy Research Network, 1180 Vienna, Austria
6   European Forest Institute, Mediterranean Facility (EFIMED), 08025 Barcelona, Spain; elena.gorriz@ctfc.es (E.G.-M.); mariana.melnykovych@ukr.net (M.M.)
7   Forest Science and Technology Center of Catalonia (CTFC), European Forest Institute, 25280 Solsona, Spain
8   Economics and Social Sciences Research Unit, Swiss Federal Research Institute for Forest, Snow and Landscape Research WSL, 8903 Birmensdorf, Switzerland
9   SEEDS-Int, Centre Ivoire, Sin el Fil 2707 5501, Lebanon; patricia.sfeir@seeds-int.org
10  FAO Regional Office for North Africa, Tunis 1000, Tunisia; arbia.labidi@fao.org (L.A.); mohammed.bengoumi@fao.org (M.B.); houda.gibson@gmail.com (H.C.)
11  Cooperative Research, Mediterranean Agronomic Institute of Zaragoza (IAMZ), 50059 Zaragoza, Spain; lucia@mallata.com
12  Social, Economic and Geographical Sciences Group, The James Hutton Institute, Aberdeen AB15 8QH, UK; maria.nijnik@hutton.ac.uk
*   Correspondence: simo.sarkki@oulu.fi

**Abstract:** Social innovations can tackle various challenges related to gender equity in rural areas, especially when such innovations are initiated and developed by women themselves. We examine cases located in rural areas of Canada, Italy, Lebanon, Morocco, and Serbia, where women are marginalized by gender roles, patriarchal values, male dominated economy and policy, and lack of opportunities for education and employment. Our objective is to analyze five case studies on how women-led social innovation processes can tackle gender equity related challenges manifested at the levels of everyday practice, institutions, and cognitive frames. The analyses are based on interviews, workshops, literature screening, and are examined via the qualitative abductive method. Results summarize challenges that rural women are facing, explore social innovation initiatives as promising solutions, and analyze their implications on gender equity in the five case studies. Based on our results we propose a new concept: reconstructive social innovation cycle. It refers to is defined as cyclical innovation processes that engage women via civil society initiatives. These initiatives reconstruct the existing state of affairs, by questioning marginalizing and discriminative practices, institutions, and cognitive frames that are often perceived as normal. The new concept helps with to assessing the implications that women-led social innovations have for gender equity.

**Keywords:** adaptive cycle; empowerment; gender equity; processes; rural women; qualitative analysis; UN SDG 5; marginalized areas

## 1. Introduction

The United Nations (UN) 2030 Agenda for Sustainable Development aims "to leave no one behind" in development. It also "endeavors to reach the furthest behind first" [1]. In many rural areas of the world, one group in risk of being left behind is women. Challenges remain despite promises by many global policies to enhance gender equity [2]. The Sustainable Development Goal (SDG) 5 addresses this challenge by aiming to "achieve gender equality and empower all women and girls." The UN SDG 5 [3] lists some of these related drivers, including "insufficient progress on structural issues at the root of gender inequality, such as legal discrimination, unfair social norms and attitudes, decision-making on sexual and reproductive issues, and low levels of political participation." Concerning the relevance of women and sustainable development, The Organization for Economic Co-operation and Development (OECD) [4] notes that "Although women account for over one-half of the potential talent base throughout the world, as a group they have been marginalized and their economic, social, and environmental contributions go in large part unrealized."

The challenge of gender equity is particularly pressing in rural areas, where patriarchal gender roles often prevail, blocking women's agency, and compromising their well-being [5,6]. A promising way to address the challenge of gender equity are social innovations [7–10]. Social innovations have been defined as civil society-based processes that meet social needs which policies or market solutions have failed to achieve, by reconfiguring attitudes and network and governance arrangements [11–14]. According to their potential impacts, social innovations have been categorized as operating on one of these three general levels: *incremental*, when they seek to change practices (e.g., social enterprises produce new goods and services), *institutional,* when they tend to change market or policy structures (e.g., new rules), or *disruptive*, when they attempt to change existing cognitive frames (e.g., to develop new values) [15,16].

By drawing on the feminist literature dealing with structural and symbolic inequality we will enrich the above conceptualizations of social innovations via two key observations on gender perspectives and gender equity. Firstly, we apply the idea of performativity by deriving it from the work of Butler [17]. Butler affirms that gender is an identity constituted and instituted through a repetition of acts, as part of processes that construct social reality via language, gestures, and symbolic social signs [17]. Thus, gender equity is not fixed in time, but emergent and constantly becoming. The concept of performativity stresses that acting differently and demonstrating the potential of alternative ways of doing things can have much bigger implications than simply changing everyday practices. It is also about questioning, challenging, acting against, and even changing the existing norms by acts that reconstruct matters underpinning gender equity.

Secondly, in order to augment knowledge on how social innovations can trigger action towards gender equality, we adapted the theory of Fraser [18] on gender justice. The theory connects gender justice to the redistribution of (1) economic resources (corresponding to redistribution struggles and classes), (2) cultural resources (corresponding to recognition and status), and (3) political resources (corresponding to framing struggles and representation) (see also Sabsay [19]). Regarding their impacts, the three types of resource redistribution in Fraser's theory come close to the three levels of social innovation impacts identified in previous social innovation literature [15,16]. Incremental social innovation relates to the redistribution of economic resources (e.g., possibilities to earn an income and receive societal opportunities, irrespective of someone's gender). Institutional social innovation largely relates to changes associated with politics (e.g., to receive equal representation in political and societal systems and have genuine possibilities to influence those systems). Disruptive social innovation requires changes at the level of cultural values, norms, and human cognitive frames (e.g., to ensure that everyone should be valued equally, irrespective of gender).

All three dimensions (i.e., incremental, institutional, and disruptive) associated with societal practices, institutions, and cognitive frames are intertwined, and necessary for

attaining gender equity. However, the discriminative, but so-called "normal" state of affairs tends to persist (e.g., via performing the prevailing gender roles in everyday practices), and thus women's possibilities to fully participate in social life and be equally recognized continue to be compromised [18].

These structural challenges with gender equity have not been adequately addressed in the social innovation literature that focuses on the reconfiguration of societal practices. Furthermore, women-led social innovations in rural areas have largely remained out of focus. The Scopus search (performed on 28 October 2020) for "Social innovation", "Women"/"Gender", and "Rural" resulted only in 15 items-out of which seven papers were on technical and medical analyses, therefore, of no relevance for the scope of our paper. The other nine papers used analytical concepts of feminist coalitions in connection to matters of care [20], social capital and entrepreneurial activities [21], women's empowerment [22], responsible micro-franchising as a place-based social innovation [23], technology interventions as social innovations to close gender-technology gaps [24], SWOT analysis [25], the role of social innovation in dispute resolution linked to marginalized women [26], and institutionalization process of gender equality and non-discrimination [27].

Thus, the papers on women-led social innovation in rural areas are scarce. Multi-case analyses from several continents are non-existent, therefore forming an empirical knowledge gap. Conceptually, none of the papers considers everyday societal practices, institutions, and cognitive frames simultaneously as interlinked bases and targets for the gender equity. In the present paper, we seek to fill these conceptual and empirical knowledge gaps.

The main objective of this paper is to analyze five case studies from Canada, Italy, Lebanon, Morocco, and Serbia–and based on them, develop theoretical concepts supported by empirical evidence. This is done to advance scientific knowledge on how rural women-led social innovations can tackle gender equity related challenges manifested at different levels (everyday practices, institutions, and human cognitive frames).

We applied a combination of participatory methods (including interviews and workshops) and literature screening on the case studies. The empirical materials were analyzed by social science techniques centered around a qualitative abductive method. The case study analysis resulted in proposing a heuristic model called *reconstructive social innovation cycle*, which is a new concept, and we believe that it can contribute to a better understanding of the effects of social innovations on gender equity.

The paper is structured as follows: first, we outline a brief empirical background on the literature available on gender equity challenges faced by rural women—and on social innovations aiming to tackle those challenges Section 2. Then, material and methods to analyze the case studies are presented Section 3. Next, we present a comprehensive analysis of the five case studies Section 4. In Section 5, we discuss the lessons learned from these case studies in order to develop a reconstructive social innovation cycle heuristic Section 6 concludes this paper.

## 2. Empirical Background: Gender Equity and Social Innovation in Rural Areas

Rural women still face serious disadvantages compared not only to rural men, but also to urban women [5,6]. women globally are "less likely to participate in the labor market than men, are more likely to be unemployed than men, and are over-represented in informal and vulnerable employment, bear disproportionate responsibility for unpaid care and domestic work, less paid than men, with the gender pay gap being estimated at 23 percent" [28]. According to The Food and Agriculture Organization of the United Nations (FAO) [29] challenges for rural women include social norms that limit women's agency, lack of education, and lack of legal, and financial entitlements. In addition, women are just 13% of agricultural land holders, and are often underrepresented in political decision making at the local level [30]. Rural gender issues also affect farm-level household relations, policy development, and identities tied to femininity and masculinity [31,32]. Živojinovic et al. [33] note that traditional and still dominant patriarchal values and practices create gendered

bias in informal institutions, thus undermining rural women's possibilities and functioning as barriers for their empowerment. Furthermore, these types of structural issues that are blocking gender equity and empowerment are frequently considered as "normal state of affairs" [34–36]. For example, cultural assumptions and values may undermine women's agency, and therefore women's efforts to engage in different types of activities (e.g., in new entrepreneurial initiatives) may be dismissed as unimportant or even as odd [33,36,37]. Thus, rural women are often in disadvantaged positions within their families and larger society when compared to men.

To cope with the above-named challenges, women-led social innovations are increasingly emerging as potential responses to the disparities, with the aim of fostering changes towards social equity and an increased women empowerment in rural areas. For instance, social innovation in the form of social entrepreneurship provides rural women with opportunities for work, empowering them by enabling creation of micro-enterprises [38], and increasing women's access to income, and motivating them to be involved in political activities [8,32]. Furthermore, via social innovations that respond to rural community needs, women are increasingly being recognized as leaders and entrepreneurs in their communities [32,38–47]. Social innovations can also help women to develop their management strategies and marketing skills for improving and expanding their businesses [38,40]. Hence, although challenges remain [48–50], with structural issues often being at the root of gender inequality, women-led social innovation initiatives have a potential to change the status quo towards gender equity. We examine five cases of women-led social innovations and analyze their potential to enhance gender equity at the level of everyday practices, institutions, and cognitive frames.

## 3. Materials and Methods

The case studies focus on social innovations that are initiated by women and aim to empower rural women. The cases are located in rural areas in Canada, Italy, Lebanon, Morocco, and Serbia. The women-led social innovations under investigation aim to increase opportunities for women, support their possibilities for agency, enhance their well-being, and reduce dependencies on male dominated structures (e.g., family, business, politics, and social hierarchies). The cases link to the themes of employment, education, identity, gender relationships and arrangements in nature-based (agricultural) practices. Our main target is to examine how women-led social innovations contribute to gender equity at the level of everyday practices, institutions, and cognitive frames. We choose the local level cases from diverse countries, continents, and cultures, assuming that integration of findings from different contexts will provide a sound basis for robust conceptualizations. The cases, representing cultural, institutional, and geographic diversity, offered strong starting points for analyzing prevailing links between social innovation and gender equity, allowing certain generalizations by going back and forth between empirical cases and literature elaborations. All cases except for the one in Canada were chosen within H2020 SIMRA project (http://www.simra-h2020.eu), by screening over 200 social innovation initiatives in Europe and Mediterranean and selecting the most suitable ones for further analysis. All together SIMRA assessed 31 cases of social innovation. The four case studies in the present paper were chosen by the criteria of major role of women in the initiatives. The Canadian case was chosen to complement SIMRA cases with a women-led social innovation example from Nunavut increasing cultural, institutional, and geographic diversity of the set of five case studies analyzed in the present paper. The interviewees, and focus group, and workshop participants within the case studies were selected based on extent of their engagement in the women-led social innovation initiatives thereby covering key actors in the initiatives enabling robust analysis.

Data were collected between the years of 2017 and 2019 through a detailed observation and field work in each of the case study, and primary data collection was complemented by media and literature screening Table 1. The interview data were transcribed, and notes were taken during fieldwork to enable the subsequent analysis. The fieldwork and data

collection methods varied across case studies, for the purpose of this paper, but each case study enabled us to build up an in-depth knowledge: (i) of the starting conditions of rural women with regard to gender inequality, (ii) of triggers of social innovations (i.e., what human needs and/or events triggered the start of social initiatives), and (iii) of outcomes and impacts of social innovations in terms of gender equity.

After the empirical data was collected, the analysis was conducted in three phases. First, the five case studies were reported via a standardized fact sheets type of format, 5–10 pages long, with a basic information on social innovations, contexts where the social innovations emerged, issues supporting the social innovations, roles of key actors and impacts of the initiatives. Second, the fact sheets were turned to narratives (see also Vercher et al. [52] and concise case descriptions, which were around 700 words long. These case narratives included the knowledge about general rural context, specific challenges for gender equity, social innovation activities and related relevant aspects, and impacts generated by the social innovation. In the third phase, the first author edited the case narratives with the information from fact sheets to come up with case descriptions of a similar structure. The case descriptions were comprised of five paragraphs: (1) problems with gender equity as triggers for social innovation, (2) social innovation initiatives, (3) impacts of the social innovation initiatives on gender equity, (4) intertwined impacts of the social innovations associated with everyday practices, institutions, and cognitive frames, and (5) a cyclical character of the social innovation initiatives. These case descriptions were then checked by authors responsible for each case study, and cross commented and edited by all authors.

The analytical methods in use were abductive rather than inductive or deductive [53,54]. Our research focuses on women-led social innovations, which offers a suitable topic for abductive analysis, because there is existing literature that theorizes social innovations [55], the changes induced with such innovations, and women's empowerment in general. However, only few sources address women's roles in social innovations together with all these other aspects, while also addressing social innovations in the rural context. Unlike inductive and deductive approaches to theoretical research designs, abductive analysis enables explorations of the intermingling empirical analysis of social innovations with conceptual views, which together can be used to enhance gender inequality. Timmermans and Tavory [56] argue that abductive reasoning should be a guiding principle of empirically based theory construction. Abductive analysis is based on an iterative dialogue between data and existing as well as novel conceptualizations [56]. Key phases of the analytical abductive process applied are summarized in Supplementary Materials.

**Table 1.** Overview of the case studies.

| Name of the Social Innovation | "Miqqut" Programs by the Ilitaqsiniq Nunavut Literacy Council (Ilitaqsiniq-NLC) | "Learning, Growing, Living with Women Farmers" Social Cooperative | "Jana Al Ayadi" Cooperative Aiming for Economic Empowerment of Women | Afoulki Cooperative of Rural Women in MOROCCO | Radanska Ruža Social Enterprise Employing Marginalized Women |
|---|---|---|---|---|---|
| Location | Nunavut, Canada | South Tyrol, Italy | Deir El Ahmar, Lebanon | South Morocco, Morocco | Municipality of Lebane, South Serbia |
| Key challenges for rural women | Intergenerational and socio-psychological traumas from colonial developments, which are connected to losses of culture and tradition, identity confusion, domestic and other types of violence, addictions and breaking down of families. Lack of support with education and childcare, self-confidence issues, lack of housing, geographical isolation, sexism in male-dominated work sectors. | Economic dependency of women on male members in farm families, gendered roles on farms and unpaid work, lack of professionalization and of a specific role of women on farms, resulting in women taking care of many different tasks from child raising to household work, which do not entail strategic decision-making powers relating to family businesses, and patriarchal value structures. | Low levels of literacy, unemployment, male dominated businesses, patriarchal value structures and gender roles, and lack of recognition of women's agency. | High unemployment and migration rates, poverty, low levels of literacy, subsistence family farming and non-wood forest products as a source of income. | Strong patriarchal gender roles and severe unemployment especially of women |
| Social innovation activities | Not-for-profit organization providing culturally relevant non-formal learning programs (e.g., in sewing) with embedded literacy and essential skills training | Social cooperative providing training program and organization of childcare service provision by women farmers on their farm according to nature pedagogy values. | Women led cooperative specialized in the production and marketing of local authentic products, employing local women. | Rural women's cooperative created with the aim to improve the livelihoods of rural women through the valorization and commercialization of the Argan oil. | Social enterprise producing traditional agricultural added-value products employing marginalized women |
| Temporal scale | 2011 | 2006 | 2005 | 2004 | 2015 |
| Empirical materials | Participant observation in 2018–2019. Semi-structured interviews on this topic ($n$ = 3), and on local women's lives from various different perspectives ($n$ = 50). Literature review (scientific literature, public/policy reports, national statistics, and grey literature). Social media screening | Semi-structured interviews ($n$ = 11). Focus group ($n$ = 1), Survey ($n$ = 21). Field observations. Literature review (see [32,46,51]). Workshop for discussing of results from previous empirical work. | Field observations (regularly in the period 2017–2019) Unstructured interviews with members of the cooperative ($n$ = 3) Participatory video; Workshop with members of the cooperative, local authorities and other stakeholders | Structured Interviews ($n$ = 5) semi-structured interview ($n$ = 5) | Semi-structured interviews ($n$ = 4). Document analysis focusing on grey literature. Media screening |

## 4. Results

### 4.1. Miqqut Program by Ilitaqsiniq-Nunavut Literacy Council ("Ilitaqsiniq-NLC")

The not-for-profit organization Ilitaqsiniq-NLC is led by local Nunavut women and offers culturally relevant non-formal learning programs (sewing, cooking, small-engine repair, etc.) with embedded literacy and essential skills training (Inuktitut and English, maths, working life skills, etc.). Both Inuit women and men in Nunavut have been impacted by colonization. Impacts include the end of camp life, migrations, and forced relocations to permanent settlements controlled by outsiders, residential schools, and hostels for children run by outsiders, medical relocations that separated children and families, and the undermining of Inuit language and culture by outsiders (see MMIWG [57] (pp. 294–324); Tester [58]; Kral [59]; Tester and Kulchyski [60]). The historical, intergenerational, and socio-psychological traumas connected to the above processes and the "loss of culture and tradition [ . . . ] and loss of control over individual and collective destiny" [61] (p. 3) have broken down families, led to identity confusion and addictions, and subjected many Inuit women to domestic and other type of violence from men [57,62,63]. (MMIWG 2019; Rotenberg 2019; Mancini & Mancini 2007). Furthermore, many Inuit women's active participation in workforce and public life continues to be hindered by, among other factors, the lack of education. There are Inuit women who have not even finished secondary education, as they have needed to look after their own children and provide for their families. There is a lack of accessible childcare, and insufficient support and engagement with schools and their curricula. All these factors are among the reasons for women to discontinue their formal education [64] (pp. 12–15); [65] (p. 17). Women's employment and other economic participation are also hindered by the lack of self-confidence and affordable housing, and due to geographical isolation of Nunavut communities [64] (p. 18), as well as because of sexism at workplaces where women are a minority (e.g., in the mining sector: see [57] (pp. 584–593).

Ilitaqsiniq-NLC was established in 1999—the same year as Nunavut became a self-governing Inuit territory. The organization's most well-known program, "Miqqut", combines sewing that draws from Inuit traditions with literacy and essential skills training. The Miqqut program began in 2011 through a pilot program with connected community research and evaluation efforts by 17 women [66,67]. Within the Miqqut program, Ilitaqsiniq-NLC has chosen to enhance Inuit women's empowerment by focusing on two key areas: non-formal education (to address the deficiencies and gaps in the Nunavut's formal education system; see Auditor General of Canada [68]; Pucci [69]; Minogue [70]; Skutnabb-Kangas et al. [71]), and cultural revitalization (by, among other activities, promoting intergenerational knowledge exchange on traditional Inuit sewing and the use of Inuktitut language during the program; Kusugak [72]. The longer Miqqut programs prioritize admitting participants who are not in school or work and provides them with a safe working space and a clear daily routine. The program shire Inuit elders and other skilful Inuit as instructors.

Already over 200 women in Nunavut have taken part in the Miqqut programs, with the completion rate being 95% [72]. Furthermore, "85% of participants who completed programs have moved on to employment or advanced [their] education/training [further]" (ibid.) Based on monitoring and interviewing done by the community researchers, there have been changes in participants' self-perception and attitude (increased confidence; knowing more about Inuit culture and traditional skills has impacted their sense of identity, etc.) [67]; [73] (p. 450); [66] (p. 29).

The case implies that the Miqqut programs cannot be seen merely as incremental social innovations that develop practical skills. Additionally, these programs open opportunities for local women to respond to some of the failures of existing education institutions. Furthermore, by promoting Inuit cultural revitalization and participants' self-esteem, this social innovation is also challenging earlier cognitive frames that outsiders have instilled through their policies and practices.

The Nunavut case highlights the cyclical character of social innovations. The Miqqut program has evolved from pilot projects into a program that is adjusted based on research findings and regular monitoring of project outcomes [66]. The program also builds on good practices and experiences of other local programs and organizations (local social innovations are helping each other to evolve), thus also staying attuned to the changing local context and needs [66] (p. 28); [73] (pp. 441–442). Finally, there have been multiple sources of funding (both public and private), which have enabled the Miqqut programs to continue beyond "one-off" funding opportunities from individual sources.

*4.2. Women Farmers Social Cooperative—South Tyrol, Italy*

The women social cooperative, "Learning, growing, living with women farmers" organizes childcare services provision on the farm according to a nature pedagogics principle. It started in 2006 with the aim to create professional opportunities for women farmers, willing or in need to remain and live in farms in remote rural valleys of South Tyrol, Italy. In this Province, the typical form of an agricultural enterprise is the family farm. Gender roles are closely linked to the fact that the identity of the German language group has developed around the principles of rurality, patriarchy, and Catholicism [74]. According to the study by Matscher et al. [75], fulltime South Tyrolean women farmers tend to accept traditional gender-differentiated division of labor, which entails unpaid tasks: they see it as their duty to do the housekeeping, to take care of their children or relatives, or to work in the garden. However, women farmers with no off-farm job or farm-based side activity find it uncomfortable to still be financially dependent on their husbands [76]. Statistics confirm the traditional patrilinear farm succession [77] and the underrepresentation of women in leading positions in agriculture. Furthermore, the separation of gender in agriculture in South Tyrol is evident through the existence of a women farmers' association since 1981 as separate from the male dominated Farmers Union of South Tyrol.

The social cooperative has developed a training program for women farmers to provide the childcare service on their farm with the required quality standards and a good work-life balance. The initiative was started by a charismatic woman, the spokesperson of women farmers in South Tyrol, in collaboration with another woman willing to found and take part in a social cooperative. From their idea, the social cooperative was founded in 2006 with the support of some members of the South Tyrolean women farmers' organization. It organizes training courses to prepare the providers of childcare service, promotes the service and organizes it in the whole region of South Tyrol.

The initiative has had empowering effects on the women farmers involved, because it enhanced changing gender roles and power dynamics in decision-making processes at the micro level (farm), at the meso level (the community) and at the provincial level. The social innovation has been highly successful also in leading to changes in the provisional law on social agriculture via a long process of interactions within the context of male dominated farming and Farmers Union. Furthermore, in transforming the role of the farm into an educational and to a meeting point of the community, the initiative has contributed to meeting the needs of working families in rural areas to have a delocalized and flexible service in the proximity and it contributes to functions of the public administration on social services provision in rural areas. Over 500 children from 0 to 4 years of age are taken care of annually through this initiative. After a decade of experience, the cooperative currently associates over 118 women providing childcare service in rural areas.

The initiative did not lead to total abandonment of patriarchal values in South Tyrolean family farms. In fact, according to Gramm et al. [32], an empowerment of women farmers is contingent to the continuity of gendered division of labor on the family farm: as in addition to their entrepreneurial activity, women farmers are responsible for housekeeping, farming tasks and caring for their own children. This results in a high workload and little free time. Therefore, women farmers "experience autonomy, within dependence" [78], it seems that social farming could be a future perspective for women farmers who want to stay in a rural environment and to feel self-realized. Again, social innovation is related

to incremental change of practices in everyday life, but it also challenges existing gender roles, and illustrates difficulties and successes of women farmers to have their voices heard in politics, and even to lead towards new developments by enhancing gender equity at the institutional level. In terms of overall cyclical development, the Italian case implies that even highly successful social innovations may lead to new kind of challenges potentially triggering different kinds of motivations to respond to these, in the future.

*4.3. Economic Empowerment of Women in Deir el Ahmar-Lebanon*

The Jana Al Ayadi cooperative was founded in 2005 and is run by women and specializes in the production and marketing of local authentic agro-food products including jams, pickles, dried fruits, vegetables preserved in oil, sweets, and delicacies. This case study is located in the area where high outmigration to cities, low levels of literacy, unemployment, and marketing challenges are some of the issues facing the women producers' in Deir El Ahmar, Lebanon. These issues are connected to the absence of policies supporting small producers and the low budget of the agricultural sector in general and of cooperatives in particular. In addition, prevailing cultural boundaries and patriarchal gender roles still hinder the development of women-based businesses, for example, by limiting possibilities for women to explore new marketing ventures outside the boundaries of their home village. Due to the inability to overcome the male dominance in business, the women are often not able to make the "one extra mile" in marketing endeavors. However, access to markets is the only way to sustain their livelihood in economic terms. Challenges for developing the business forward include the lack of know-how, support and self confidence in their skills and their achievements.

The women-led social innovation, Jana Al Ayadi cooperative, aims to secure stable income for the women. The cooperative got support from the Social Economic and Environment Development Services (SEEDS-Int), a Lebanese NGO, arranging capacity building for the women, increasing their technical expertise in improving their marketing and boosting their sales. In addition, a new branding strategy and the establishment of a social online marketplace for selling and promoting their products was done with support from an the H2020 SIMRA project (http://www.simra-h2020.eu). The cooperative provides value for both producers and consumers by offering a marketplace for buying traditional products and the possibility for customers to get to know the story of every producer. This has helped women to find ways to improve the economic situation of their families and enhance their well-being.

In the Lebanese case, the women have learned to manage and develop their cooperative business striving to secure a sustainable model for improving their livelihoods. The women based cooperative "Jana Al Ayadi" benefited from uplifting their brand, re-designing their label, focusing the marketing efforts on star products, and creating the e-commerce platform (https://ishoprural.com), which has increased their revenues and profit. Furthermore, extension of the business enabled smallholder producers to gain a fair space and the opportunity to get in contact with the consumers, improve visibility, and boost revenues and most importantly ensure sustainable business solutions.

While the case is about incremental change of everyday practices and increasing women's access to markets, it is also very much about questioning existing gender norms in the village and local markets. The success of this cooperative as women-led business is demonstrating that women can be active in providing benefits not only for themselves, but also to the surrounding community.

Cyclical processes are reflected in the remaining difficulties of expanding the women-led business to neighboring areas. The success in their home village has been demonstrated, but in adjacent areas social boundaries of male dominated markets and patriarchal cognitive frames remain. This illustrates a cyclical relationship between demonstrating success and overcoming prejudices towards women-led business.

*4.4. Argan Co-Operative of Rural Women, Morocco*

The Afoulki umbrella cooperative aims to improve the livelihoods of rural women through the valorization and commercialization of Argan tree products. The organization is based in marginalized rural area in Morocco's region Souss Massa, which is characterized by high poverty, unemployment rates and outmigration [42]. Gender inequality adds additional challenges, further deepening the marginalization of the region. Women are mainly unpaid domestic workers and self-employed in subsistence farming. Literacy rate for men is much higher (70.3%), compared to women (42.5%), leading to men's influence over women and control of their decisions in all social spheres. Women in the region have been disadvantaged socially, educationally, politically, and economically. Thus, the triggers for establishing the cooperative were the high rate of rural women unemployment, lack of their financial independence, as well as low literacy of women, patriarchal societal norms with male-dominated social system wherein males hold authority and power, and women are mainly working in unpaid care jobs at home.

The above-mentioned challenges led to establishing the cooperative in 2004 by a local rural woman. The cooperative brought together 30 women-members who jointly started crushing argan fruits and producing artisanal argan oil. The area with its unique climate conditions is highly populated by the endemic argan tree species (*Argania spinosa*), which allows local women to make additional income by making oil out of its fruits. The aim of the Afoulki cooperative is to improve the living conditions of women by providing an enabling environment to gain a personal income and training opportunities, autonomy, and independence and to ease two major problems in the territory: women's poor socio-economic situation and the undervaluation of argan products. The cooperative was established as a result of continuing governmental efforts over the last few decades boosting the establishment of small-scale professional organizations, including cooperatives. The Afoulki cooperative was founded in frame of this initiative and been supported financially and technically by national institutions and international organizations (e.g., Ministry of Agriculture, NGOs). At the beginning, many were skeptical about the idea of creating a cooperative. This challenge was overcome through the advocacy work of the female president of the cooperative who convinced men to let their wives have a job outside the house. A significant number of local women looking for a job and willing to earn money to help sustaining their families, also financially, showed their interest in working in the cooperative.

The cooperative started with 30 women-members and expanded within the territory to include various other branches. In 2018, the cooperative enlarged and became an umbrella network for several other small cooperatives, which were established in 2018. Thus, Afoulki umbrella cooperative created the opportunity to make income for many rural women in Souss Massa Region. Nowadays, Afoulki umbrella network has more than 300 female members spread over 13 affiliated cooperatives. The members are producers and exporters of Argan oil and other Argan cosmetic products (soap, cream, etc.). The cooperatives produce different organic products (cosmetic Argan oil, culinary Argan oil, Argan soap, honey, Apricot oil, Almond oil, and Cactus oil), many of which are exported to European and American countries. The Afoulki network of cooperatives is benefiting from a strong set of connections including with its members, employees, funders, government representatives, banks, insurance companies, inter-professional federations, national agencies, and private companies. The Afoulki umbrella cooperative as a social innovation has brought together actors who were not used to working together before and helped in concentrating their efforts towards providing opportunities for rural women and enhancing environmental sustainability. Specific impacts on rural women have been their improved technical expertise and know-how enabling production mechanization of larger quantities of the products. The Afoulki umbrella cooperative also contributed to improving rural women's social capital through their increased self-confidence and creating new networks as well as through reducing inequalities within the region.

The work of the initiative is not only about employing women. Because of the trust and respect shown by the local actors ranging from the local authorities to the president of the cooperative, family dynamics have changed as men have gradually accepted their wives being part of the cooperative with well-defined work schedules. This highlights that enhancing women employment is linked to traditional gender roles, and patriarchal values. In addition, with the gained revenues, the cooperative organized many capacity building seminars for mainly female participants leading to increased expertise and know-how for women to work in the cooperative and beyond. The cooperative has had impact through educational sessions for illiterate women and by arranging care services for children, thus enabling women to work. The changes in everyday practices of women working in the cooperative have also led to increased respect and self-confidence and enabled participation of women in local decision-making processes.

An example of the cyclical characteristics of the Afoulki case is that the respect, trust, and belief in the women who initiated the cooperative made other women believe in the success as well, thus reinforcing and expanding the positive impacts along the way. Moreover, the leaders' motivation and high ambitions to develop the rural community and improve the situation of its inhabitants created self-confidence among local women on their potential to work outside their homes. Therefore, even though not all the gender equity issues have been solved, the women currently perceive themselves to have more opportunities than was the case before the Afoulki initiative.

*4.5. Radanska Ruža Social Enterprise, Serbia*

The Radanska Ruža is a social enterprise founded in 2015. It is located in Southern Serbia, in the municipality of Lebane (20,000 inhabitants), where 60% population is unemployed. Per capita income is only 130 euros per month. Elderly people, people with infirmities, single parents, and women can hardly earn the bare minimum by working in their yards and gardens. In addition to the very hard economic conditions, very traditional and a (still) dominant patriarchal value and gender role system exists in Serbia, where women have often been excluded from participation in economic activities, have "subordinate role" in the family, and lack of access to property rights [33].

Radanska Ruža is a non-profit limited liability company, and a public-private partnership between "Women's Association Ruža" and the municipality. This social enterprise is securing employment for women from vulnerable groups, especially women with disabilities, victims of domestic violence and single mothers. It collaborates with local agricultural producers to assure the availability of raw materials for the production of natural vegetable and fruit value added products based on traditional recipes and by traditional hand-made techniques. It targets especially disadvantaged and marginalized women, who find not only work and income at Radanska Ruža, but also a particularly appreciative social environment.

The number of employed women has grown from 5 in 2015 to 32 in 2020. Indirectly, it engages and provides income for up to 80 households. In its work Radanska Ruža operates with a network of local producers and households for raw materials, who sometimes do the first processing step. Therefore, Radanska Ruža secures income and earnings also for other local people, in addition to the women employed by the social enterprise.

For many of the employed women, Radanska Ruža provided their first job outside their homes. Subsidies are scarcely available, and the municipality is rather inactive in supporting the initiative, even though Radanska Ruža is by far the largest employer of people with disabilities. The social innovation may at first sight seem to take place in the incremental level by providing possibilities for income for marginalized women. However, the case also links to challenging of patriarchal values, which have hindered women's employment outside their houses. Furthermore, the lack of existing subsidy system has motivated the initiative to search partners from abroad to secure jobs for women. Therefore, the case is also linked to institutional innovations. For example, in 2020, BioBalkan launched

a crowdfunding campaign in Austria with the aim to secure funding for one year of work for women.

This initiative seeks to provide continuous work for marginalized women. Thus, the social innovation initiative seeks not only to achieve impact at a certain point in time, but to maintain the positive effects for a longer period of time. However, some difficulties have occurred. For example, Radanska Ruža was a victim of the irresponsible business of a distributor who had tricked them and brought the company to brink of existence. This illustrates very insecure and unfavorable conditions that such small businesses currently face in Serbia. This also implies that social innovation may experience surprising difficulties. The cooperative survived thanks to their foreign partner BioBalkan who organized a crowdfunding campaign in Austria. Additionally, two crowdfunding campaigns were organized in Serbia. These campaigns enabled continuance of the cooperative, necessary purchasing of raw materials, and payment of salaries and taxes. Together, the results from our analysis of this social innovation initiative suggest that surprising obstacles and solutions may emerge, and that impacts of social innovations occur in a cyclical, rather than linear, manner.

### 4.6. Summary of the Findings from the Case Studies

In Table 2 we provide summary findings from the case studies regarding the strengths, weaknesses, and unique aspects of linked to each case study. We also identify cross-scale findings common for all the case studies.

**Table 2.** Summary of the findings from the case studies.

| The Social Innovation | Strengths | Weaknesses | Unique Aspects |
|---|---|---|---|
| "Miqqut" programs by the Ilitaqsiniq Nunavut Literacy Council (Ilitaqsiniq-NLC) | Opening opportunities for rural women to overcome marginalizing institutional settings in education and enhancing employment opportunities. | Initiatives not self-sufficient, but dependent on project-based funding | Connecting modern literacy with reviving cultural traditions to empower marginalized women and enhance intergenerational connectivity. Monitoring the impacts of the program regularly |
| "Learning, growing, living with women farmers" social cooperative | Initiating the process of policy-making on social agriculture in the region; empowering women farmers in the decision-making process of the family farm, as their activity becomes part of the business strategy; increasing their awareness of doing a visible and valued job for the community | In some cases, women farmers are responsible for housekeeping, farming tasks and caring for their own children in addition to their new entrepreneurial activity. The case can be considered a clear example of the ongoing tension between farming tradition and modernization in mountain territories | Changing gender roles in decision-making processes at the micro level (farm), at the meso level (the community) and the provincial level |
| "Jana Al Ayadi" cooperative aiming for economic empowerment of women | Creating successful business employing marginalized women by enterprise producing and marketing agro-food products despite strong patriarchal gender roles | The success in the home village of the cooperative has been demonstrated, but in adjacent areas, social boundaries of male dominated markets and patriarchal cognitive frames remain | Extension of the business by the cooperative enabled smallholder producers to gain a fair space for marketing their products and connecting with the consumers, improve visibility and boost revenues |
| Afoulki cooperative of rural women in Morocco | Improving the living conditions of women by providing an enabling environment to gain a personal income and training opportunities, autonomy, and independence | Women are still mainly unpaid domestic workers and self-employed in subsistence farming | In 2018, the cooperative enlarged and became an umbrella network, for several other small cooperatives |

**Table 2.** *Cont.*

| The Social Innovation | Strengths | Weaknesses | Unique Aspects |
|---|---|---|---|
| Radanska Ruža social enterprise employing marginalized women | It targets especially disadvantaged and marginalized women, who find not only work and income at Radanska Ruža, but also a particularly appreciative social environment. | The initiative may be vulnerable due to high reliance on individual key actor | During an unexpected crisis, several crowdfunding campaigns were organized enabling continuance of the social enterprise |
| Cross-case findings | • Social innovations play a key role in enhancing self-esteem and self-confidence of participants leading to further capacity to act (e.g., economy, education, politics). <br> • Individual innovative women have played key roles in establishing, maintaining and extending the initiatives. <br> • Women led social innovation initiatives do solve some but not all challenges regarding gender equity. <br> • Women led enterprises and initiatives are often met with suspicion by males and external society, and therefore women led initiatives need to overcome prejudices and deal with discriminative gender roles, institutions, and cognitive frames. | | |

## 5. Discussion

### 5.1. Intertwined Incremental, Institutional, and Disruptive Dimensions of Social Innovation

The cases of women-led social innovations analyzed above highlight that social innovations seem at first sight to be taking place at the level of everyday practices. However, closer examination reveals that the everyday practices are linked to institutions, cognitive frames, and value hierarchies. This is because it is impossible to initiate new everyday practices without also interfering into the underpinning structural challenges (e.g., patriarchal values; biased gender roles; male dominated politics and economy) in which the problems for gender equity are rooted. This finding reifies the view of the feminist theories on the relevance of performativity and the linkages of everyday practices to structural challenges for gender equity [17,18]. Therefore, the aims and impacts of women-led social innovations do not properly fit into the categorizations of the existing social innovation literature which treat impacts as either *Incremental, Institutional or Disruptive*. Previous literature has proposed a linear social innovation continuum. According to this approach, incremental, institutional, and disruptive social innovations vary in their scalability and impact, which grow when moving from incremental via institutional to disruptive [16]. Based on our five case studies from different geographical, institutional, and cultural contexts, we propose an alternative conceptualization to understand the interlinked impacts of social innovations. Among other things, our heuristic considers that individual women-led social innovations can cover all three levels of impacts—and these processes are often cyclical rather than linear. We call this new conceptualization a *reconstructive social innovation cycle* Figure 1.

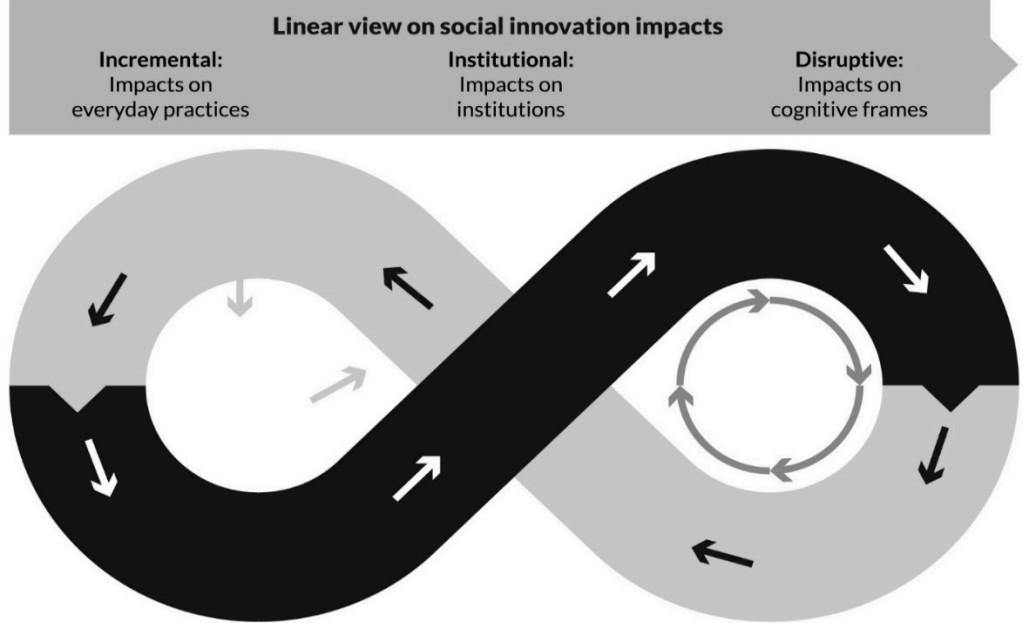

**Figure 1.** Two alternatives to consider impacts of social innovation. Firstly, according to a linear view, impacts grow in magnitude from Incremental via Institutional to Disruptive. Secondly, according to the proposed reconstructive social innovation cycle heuristic, the different levels of impacts are intertwined and inseparable. We propose that for understanding the impacts of women-led social innovations on gender equity the reconstructive social innovation cycle is more appropriate.

### 5.2. Reconstructive Social Innovation Cycles

Our results indicate that rather than producing impacts in a linear way, social innovations in the women-led cases analyzed develop through cyclical processes. This specifically concerns project-based social innovations (e.g., "Miqqut" programs). Social innovations are not likely to resolve all the problems they are targeting, thus leading to the conclusion that the end is just a new beginning, with different, and often improved situation, but still with remaining challenges calling for novel ways to enhance gender equity (e.g., "Learning, growing, living with women farmers" social cooperative). A continuous interplay between successes of women-led social innovation and (overcoming) discriminative prejudices (e.g., economy and business as male dominated sphere of life) likely explains effects of evolving social innovations on gender equity (e.g., "Jana Al Ayadi" cooperative). Women-led social innovations help women to gain more resources and self-confidence, leading to a renewal of the initiatives and possible further impacts ("Afoulki" cooperative). Social innovations aim to maintain the positive effects in the long-term, and to do so in changing situations, implying that impacts are not stable, and that the impacts can be considered also as continuous processes (e.g., "Radanska Ruža" social enterprise). In social innovation literature, evolvement of social innovations has been considered by identifying Divergent Development Paths [14], that can lead to reconfiguration of social practices and institutional changes [13]. In addition, pursuing certain values can trigger social innovation, and novel practices by social innovation may lead to subsequent value changes thereby changing the initial triggers and context which motivates future social innovations [79]. Building on these literatures and based on the case studies, we propose the notion of reconstructive social innovation cycle to provide an alternative way of understanding the impacts of women-led social innovations in the localities where they emerge and evolve.

A *reconstructive social innovation cycle* is defined as: a cyclical process of engaging women via civil society initiatives that are reframing the existing state of affairs by chang-

ing or questioning marginalizing and discriminative practices, institutions and cognitive frames that are often perceived as normal. The "reconstructive social innovation cycle" builds on from the concept of adaptive cycle [80–82], and the results of the case studies analysis carried out. We chose the adaptive cycle as a point of elaboration because it aims at understanding dynamics and persistence of changes and conceptualizes changes as cyclical processes [80], and it has been applied also to examine social innovations [83,84].

An adaptive cycle consists of four phases: (1) growth or exploitation, (2) conservation, (3) collapse or release, and (4) reorganization. The (1) growth phase is about slow growth of a system, (2) conservation phase is when a system has matured and sustained over a certain period of time. In the (3) release phase, the system collapses in its existing state, after which the system is rapidly (4) reorganizing itself and enabling to grow either in the same or a different configuration or a systemic state [80,81]. In the adaptive cycle, there are two types of transformations. The first is the so-called "back loop", which is a rapid phase of system's reorganization after collapse leading to a renewal. The second is the so-called "front loop" which is characterized as a slow change of growth and accumulation, taking place in and between the growth and conservation phases [80,82]. Based on our case studies and the adaptive cycle theory, we propose the reconstructive social innovation cycle Figure 2 to better understand women-led social innovation processes and their impacts on gender equity-which is detailed in the next subsections. Next, we propose empirically based novel concepts to understand the cyclical processes of women-led social innovations.

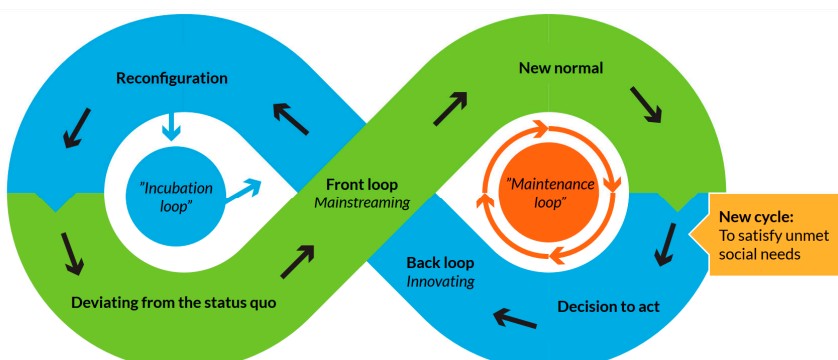

**Figure 2.** Reconstructive social innovation cycle to understand implications of women-led social innovations on gender equity, in situations where gender equity has been compromised by the normalization of discriminative practices, institutions and cognitive frames.

### 5.2.1. Decisions to Act Leading to Reconfiguration

In the adaptive cycle, the change starts by a rapid release phase, which is about letting go, for example, of values, priorities and dependencies that drive a certain state of affairs, or a systemic state. Differently, in the reconstructive social innovation cycle, we propose that "release" requires active decisions to act for change. Our cases highlighted different unmet social needs resulting from male dominated identities, values, practices and economic sphere of life, and lack of opportunities for education and employment, especially for women, in rural areas. These problems for gender equity were tackled by decisions to act and e.g., to start training courses combining traditional knowledge and skills with the capacity building in modern literacy ("Miqqut" programs); by training women farmers in providing childcare services to reorganize the roles on farms which entail a stronger negotiation power for women on the family business ("Learning, growing, living with women farmers" social cooperative), and by establishing social enterprises employing especially marginalized women ("Jana Al Ayadi" cooperative; "Radanska Ruža" social enterprise), establishing a cooperative to enable women work also out of their homes ("Afoulki" cooperative). All these examples were based on self-organization of women to improve the lives for themselves and their peers, having often one or few key actors who started the process of change. According to Eleutério and Van Amstel [20]

feminist coalitions and collaborations emerge around shared visions to combat historical and cultural problems, even if they are considered difficult to solve. The cases highlight women-led social innovation initiatives as form of self-organization and as means to combat challenges related to social equity. The social innovation literature typically refers to such initiatives as reconfigurations of social practices [12]. According to Lindberg et al. [7] (p.472), "gendered social innovation encompasses the identification of unsolved societal challenges of gender inequality and unmet needs among women or men as underrepresented or disadvantaged groups in various areas, motivating the development of new solutions by inclusive innovation processes." The decisions to act led to establishment of the gendered social innovation processes to combat discriminative institutions and norms manifested also at the level of everyday practices.

### 5.2.2. Deviating from the Status Quo Consisting of Discriminative Institutions and Norms

A change from the adaptive cycle terminology of "growth phase" to "deviating from the status quo" highlights specific challenges linked to gender equity as observed in the case studies. "Deviating from the status quo" can be understood as doing things differently than the prevailing and discriminative institutions and cognitive frames would expect. This is associated with observations that everyday practices also relate to deeper institutional and cultural constructs. Therefore, the so-called "normal" state, but which may undermine gender equity, can be challenged, and questioned by "going" against the stream. The examined cases highlighted the efforts of women to break traditional patterns of gender roles, economy, and education. This requires active efforts and self-confidence to act outside (and probably against) what is perceived as normal and accept that the steps taken and the changes observed at first might not be very big and prominent, but are representing alternatives to enhance gender equity, also at the institutional and cognitive levels. The study cases highlighted that all women-led social innovation initiatives emerged in a situation where prejudices were blocking possibilities for women, for example, by prevailing patriarchal gender roles, domination of males in spheres of local politics and economy, and the lack of employment and education opportunities for women. Therefore, the women-led social innovations had to deviate from the discriminating and marginalizing status quo.

### 5.2.3. Impacts: New Normal

If social innovations are successful, they may produce a new normal state within the locality or niche where they operate. A new normal state of affairs consists of social practices that were previously marginal or even non-existent. Arriving at the new normal may be hindered by prevailing values and decision-making processes and their impacts on one's identity. New normal is not only about changing everyday practices, but also about questioning, challenging, acting against, and even changing the existing norms by acts that reconstruct issues underpinning gender equity (see Butler [17]). In the "Miqqut" case, many women have ended up returning to employment or training or education programs, which represent new opportunities for marginalized women in Nunavut. This reflects findings on ability of social innovations help in empowering women in marginalized indigenous communities [8]. In the social cooperative of "Learning, growing, living with women farmers" more than 100 women have been trained and started working as childminders. As the collaborators of the social cooperative took over administrative tasks and the coordination of demand and supply, women became able to concentrate on their care-taking activities. South Tyrolean women farmers offering childcare services became responsible for their own entrepreneurial activity and by acquiring the access to resources. In the "Jana Al Ayadi" cooperative, the new normal state increased the self-confidence and power of women in business and in the village level decision-making. Women have become recognized in a formerly male-dominant culture when they started making economic profits and adding earnings to their family budgets, also revitalizing local economy. In the "Afoulki" cooperative case, new normal was manifested by the fact that

women have become allowed to work outside their homes and gained a degree of financial independence, instead of being fully reliant on men in their families. In the "Radanska Ruža" social enterprise case, new normal is reflected by dozens of once marginalized women having now employment opportunities in the locality with a high unemployment rate, and where formerly the women were not seen as economic actors. As a result of the initiative, many women opened their own bank accounts for the first time. Therefore, the concept of *new normal* produced by the social innovations can be used to conceptualize the impacts of social innovation initiatives promoting gender equity and empowering marginalized women. In the cases this happened for example via enhanced economic, employment and education opportunities. Therefore, these results complement findings of Maestripieri [9], who shows that participation in social innovation does protect households from worsening economic conditions. However, social innovations are seldom able to resolve all the problems they were designed to tackle. This explains the motivations to continue the initiatives beyond a periodical scope and may also motivate completely novel social innovations to address the prevailing problems by starting a new reconstructive social innovation cycle.

5.2.4. Incubation Loop and Maintenance Loop

Based on the case study analyses we propose two sub-loops to reconstructive social innovation cycle. Firstly, many innovative ideas do not get mainstreamed, but get stuck in the "Incubation loop" where key actors collaborate and are innovating, but the ideas remain in small circles and do not get wider attention. The challenge of getting stuck in the "Incubation loop" was broken in the case of "Miqqut" programs. Ilitaqsiniq-NLC that runs the Miqqut programs has been very successful in year after year acquiring grants and sponsorships for its programs, which has enabled the programs to grow and evolve from sewing to also other skills (e.g., cooking and woodwork) and new target groups. The "Learning, growing, living with women farmers" social cooperative case managed to overcome the incubation loop, where women farmers were skeptical about starting to provide the service of childcare on the farm. The social cooperative, which showed to the farmers the feasibility of the business and the benefits it gave to women in terms of professionalization, economic independence, self-realization, their role on the farm, and a changed role of the farm in the community. The "Jana Al Ayadi" cooperative case broke the incubation loop by emerging self-confidence of the involved women and by proving that women-led business can flourish and provide benefits for the community. In Radanska Ruža social enterprise case the incubation loop was eventually broken after a challenging period of time while moving from a grant-based financing to a self-sustained initiative and reaching the economic viability.

The other sub-loop in a reconstructive social innovation cycle is called "maintenance loop", which points to the need of sustaining new practices and the changed previous "normal" state of affairs initiated by the social innovations. The "new normal" may not be self-sustaining without the continuance of social innovations. For example, in the "Miqqut" programs case, the innovators monitored the results and fine-tuned the program accordingly. They also created new programs to be able to engage with a wider range of local people (not only marginalized women) and secure more funding. In the "Learning, growing, living with women farmers" social cooperative case, changes in the provincial law and enhancing women's roles in the Farmers Union can be considered as new developments that help maintaining the new normal. In the "Radanska Ruža" social enterprise case, active search for new funders and business partners aims to secure positive impacts by maintaining the initiative in the long term. In the cases of "Jana Al Ayadi" cooperative and the "Afoulki" cooperative, economic success is helping women to maintain the new normal through financing, but also by creating respect towards women and their self-confidence in maintaining the business running also in the future. Therefore, institutionalization processes enhancing gender equity and combating discrimination are important aspects of

social innovations for creating sustained grounds for subsequent women empowerment in rural areas [27].

## 6. Conclusions

This paper started by two key observations. Firstly, everyday practices are linked to deeper institutional structures and cognitive frames. Therefore, social innovations seeking to enhance gender equity do not fit well into the earlier classifications of social innovations and their impacts as being either Incremental, Institutional or Disruptive. Secondly, the existing "normal" state of affairs often discriminates and marginalizes women and is reified or challenged in continuous social innovation processes. Hence, social innovations seeking to enhance gender equity are better understood as cyclical rather than linear processes. In addition, treating social innovations and their impacts as if they were frozen in time neglects the processual nature of the continuously evolving reality, where gender equities are (re)constructed again and again. Insights from the feminist literature on performativity [17] and gender justice [18] provided significant help in analyzing the women-led social innovations. The feminist theories highlight structural challenges for gender equity and the women-led social innovations are exemplars of potential solutions to ease these challenges. Therefore, for future research, we propose further applications of the diverse feminist literature to examine social innovations. In the present paper, we have proposed a heuristic framework to better understand relationships between women-led social innovations and gender equity. Further studies are needed to apply and test the proposed heuristics in other cases. Limitations of the present paper relate to relatively small number of case studies on which the theoretical conclusions were based.

In the present paper, the combination of having the feminist theories at the background and in-depth case studies on women-led social innovation as the empirical material enabled us to propose a novel heuristic called reconstructive social innovation cycle, which can in particular enhance the ability to assess implications of women-led social innovations for gender equity. The heuristics recognizes that perceived "normal" state of affairs may be marginalizing and discriminating women. In order to reconstruct a new normal, the social innovations seeking to enhance gender equity need to challenge the prevailing and often fundamental social, cultural, political/institutional, and economic structures by demonstrating that alternatives are possible and viable. Such initiatives need to deviate from the status quo to break the normalized discriminative or marginalizing conventions. In addition, the heuristics acknowledges that the women-led social innovations need to actively maintain the new more equitable normal state of affairs. The cases examined here showed a great promise that women-led social innovation can really change gender inequality conditions in rural areas towards "new normalities" entailing more gender equity and enacting rural women to contribute to rural development. Despite extensive success, the women-led social innovations are very unlikely to resolve all the challenges for gender equity. Therefore, the social innovations aiming to enhance gender equity can be understood as cyclical processes, where subsequent initiatives can make progress starting from the previous social innovations towards sustainability.

**Supplementary Materials:** The following are available online at https://www.mdpi.com/2071-1050/13/3/1231/s1, Table S1: Seven step abductive analysis method applied in the present paper.

**Author Contributions:** S.S. led the writing process, synthetized the case studies and developed the overall argument. C.D.T. commented and edited the paper in many phases, provided South Tyrol case study and was responsible for the call for fact sheets on the case studies. J.F. provided the Nunavut case study, supplied insights from feminist literature, and commented on and edited the paper in its various phases. I.Ž. provided the case study from south Serbia, and commented on the paper in various phases. A.L. commented and contributed to writing the paper in various phases, and supplied insights on gender equity and rural development. E.G.-M. commented and contributed to writing up the paper in various phases. M.M. synthetized materials on the South Moroccan case, supplied insights on gender equity in rural areas, and commented and contributed to the paper in its various phases. P.R.S., L.A., M.B., H.C., V.G., and L.L.M. helped in developing

the case descriptions. E.R. commented on the paper. M.N. gave general comments on the overall argument and did scientific editing and final proofing for the paper. All authors have read and agreed to the published version of the manuscript.

**Funding:** The authors are grateful to the European Commission for financial support to the project on Social Innovation in Marginalised Rural Areas (SIMRA) provided from the European Union's Horizon 2020 research and innovation programme under Grant Agreement No 677622. Contribution of the author from the James Hutton Institute to this study was also partly funded by the Rural & Environment Science & Analytical Services Division of the Scottish Government through its Strategic Research Programme (2016–2021). The Nunavut/Canada case study was supported by Nordforsk Nordic Centre of Excellence: Resource Extraction and Sustainable Arctic Communities (REXSAC: project number 76938).

**Institutional Review Board Statement:** The case studies located in Italy, Lebanon, Morocco, and Serbia were conducted in the frame of Horizon 2020 project SIMRA, and the research design complies with the legal guidelines on research ethics. The ethical clearance procedure is described in the SIMRA Deliverable 5.1 (http://www.simra-h2020.eu/wp-content/uploads/2018/06/SIMRA-D5.1 _Case-Study-Protocols-and-Final-Synthetic-Description-for-Each-Case-Study-1-1.pdf). Regarding the Canadian case study Nunavut Research Institute (NRI) Research License (No. 03 027 18N-M) and University of Alberta Research Ethics Board 1 approval (Study ID: Pro00084782), were acquired.

**Informed Consent Statement:** Informed consent was obtained from all subjects involved in the study.

**Data Availability Statement:** Not Applicable.

**Acknowledgments:** The authors sincerely thank those interviewed or who otherwise participated into the case studies.

**Conflicts of Interest:** The authors declare no conflict of interest.

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
