# Peer review of "Reconstructive Social Innovation Cycles in Women-Led Initiatives in Rural Areas"

_sustainability, doi:10.3390/su13031231_

Round 1

Reviewer 1 Report

Thanks to the authors for their work, this is an interesting and wide-ranging investigation. However, I must make a few brief recommendations for its publication.   1.- Abstract: I would like the authors to introduce the most relevant results in this section and conclusions.    2.- The quotations are not in the required format. The mdpi journals require that they be numbered in the text and placed in the bibliography section according to text order and not alphabetical order. Please take other journal articles as examples.   3.- Introduction: Although this section could be more extensive, it is well resolved. Authors should ensure that the citations that now appear in the introduction as background to their research, appear later in the discussion. This is necessary.   4.- Materiales y métodos: Esta sección debe ser reelaborada. El primer párrafo de esta sección no es necesario. Los objetivos y la importancia del estudio ya han sido definidos previamente y, en cualquier caso, los objetivos e hipótesis deben aparecer pero al final de la introducción. Echo de menos una sección clara que describe la muestra que fue parte de cada caso de estudio y también cuáles fueron los criterios para estudiar esos 5 casos y no otros.   5.- Resultados: Se realizan análisis de los programas desarrollados de forma descriptiva. Me parece necesario que, de una manera gráfica, los autores hagan una tabla diagnóstica donde se puedan observar las fortalezas de cada programa, sus debilidades y de alguna manera establezcan comparaciones.   6. - Debate y conclusiones: son atractivas y bien elaboradas. Asegúrese de dejar claras las implicaciones teóricas y prácticas de este estudio y sus limitaciones y perspectivas.  

Author Response

Dear editor,

Please note that we have changed authorship. The second last author, Diana E. Valero López, dropped out. We hope to send the authorship change form and confirmations by all authors via email. Hope this is fine for the journal.

Sincerely, corresponding author

Dear reviewer, 

Thank you for your insightful comments, even though some of them were in foreign language. Please find our responses below.

Reviewer 1

1.- Abstract: I would like the authors to introduce the most relevant results in this section and conclusions.    2.- The quotations are not in the required format. The mdpi journals require that they be numbered in the text and placed in the bibliography section according to text order and not alphabetical order. Please take other journal articles as examples.   3.- Introduction: Although this section could be more extensive, it is well resolved. Authors should ensure that the citations that now appear in the introduction as background to their research, appear later in the discussion. This is necessary.

Response:

1: ABstract: we highlight in revised version that the proposed resconstructive social innovation cycle is based on the case studies, and hence it is our major conclusion. The word limit for abstract is 200 words and as we examine five case studies we decided not to present detailed results, but rather the concept of reconstructive social innovation cycles highlilghting that it can be used in subsequent work to assess implications of women led social innovations on gender equity.

We added several citations to discussion to make connections to literature explicit.  

Reviewer 1: References are not correctly formatted.

Response: We modified references according to instructions of the journal.

Reviewer 1:

Materials and methods: This section should be reworked. The first paragraph of this section is not necessary. The objectives and the importance of the study have already been defined previously and, in any case, the objectives and hypotheses should appear but at the end of the introduction. I miss a clear section describing the sample that was part of each case study and also what were the criteria for studying those 5 cases and not others.

Response: We added text explaining sample and case study selection to first paragraph of Material and methods. We retained the first paragraph for clarity and introducing briefly what the cases are about. Introduction presents our research questions and objectives. IN material and methods we feel constructive to explicitly link the material and methods to objectives, hence little repetition between intro and material and methdos.

Reviewer 1:

Results: The programmes developed are analysed in a descriptive way. It seems to me necessary that, in a graphic way, the authors make a diagnostic table where the strengths of each programme can be observed, its weaknesses and somehow establish comparisons. 

Response: We added a summary table at the end of results section.

Reviewer 1

Discussion and conclusions: they are attractive and well thought out. Be sure to make clear the theoretical and practical implications of this study and its limitations and perspectives. 

Response : We added some sentences to conclusion on limitations of the paper.

Reviewer 2 Report

  • Content:
  • 1- The article is very interesting and provides recent literature about the project. The theories selected of Butler and Fraser do provide the framework of the initiative, however these theories are not integrated within the data analysis, they are lingering in the introduction while they should be applied on the case studies. For example, in 5.2.3. Impacts: New normal, Butler’s gender and performativity theory should be applied to support the argument that these initiatives facilitated a change in the normative gender codes of these rural areas. 
  • 2-Fraser’s also needed to be integrated within the analysis and the description of these initiatives. 
  • Aademic Styel:
    • No need to write the quotations in italics.
    • wrong website citation: (see https://www.unwomen.org/en), in-text citation should be changed, https://www.unwomen.org/en
    • Table No. 1 is not clear, it would be better if it is restructured in an acceptable table format, it is chaotic and unclear
    • The methodology is very clear, no need to create table 2, it is an aux.
    • 51 OECD, FAO 145 who is this body? The full name of the institution should comenbefore the acronym.
  • Grammar:
  • 1-The article needs another proofreading, there are several grammatical and structural mistakes to amend. Some are mentioned bellow:
  • 44  one group  in risk of being left behind  “is” women
  • 47-48 The UN lists some of ‘these’  related drivers including
  • 50 In relation to the relevance of women ‘and’ sustainable development,
  • 221-222 These case descriptions were then checked by authors responsible for each case study, and cross commented and edited by all authors.
  • 225 The research focus(es) on women-led social
  • 2- some sentences are not clear:
  • economic production employing women.
  • 513-514 was a victim of the irresponsible business of a distributor who had tricked them and brought the company to brink of ex- istence.
  • 567 trigger social innovation feeding back to value changes

Author Response

Dear reviewer 2,

Thank you for your comments. Below we outline our reactions to your valuable comments.

Best wishes,

Authors

Aademic Style:

    No need to write the quotations in italics.

    wrong website citation: (see https://www.unwomen.org/en), in-text citation should be changed, https://www.unwomen.org/en

    Table No. 1 is not clear, it would be better if it is restructured in an acceptable table format, it is chaotic and unclear

    The methodology is very clear, no need to create table 2, it is an aux.

    51 OECD, FAO 145 who is this body? The full name of the institution should come before the acronym.

Response:

We reformatted references and table 1. We moved table 2 to annex as proposed by the reviewer.

We clarified the acronyms.

Reviewer 2

Grammar:

1-The article needs another proofreading, there are several grammatical and structural mistakes to amend. Some are mentioned bellow:

44  one group  in risk of being left behind  “is” women

47-48 The UN lists some of ‘these’  related drivers including

50 In relation to the relevance of women ‘and’ sustainable development,

221-222 These case descriptions were then checked by authors responsible for each case study, and cross commented and edited by all authors.

225 The research focus(es) on women-led social

2- some sentences are not clear:

economic production employing women.

513-514 was a victim of the irresponsible business of a distributor who had tricked them and brought the company to brink of ex- istence.

567 trigger social innovation feeding back to value changes

Response

We checked grammar.

Round 2

Reviewer 1 Report

Gracias por hacer los cambios sugeridos. Acojo con beneplácito el artículo para su publicación. En futuras ocasiones usted debe ser más completo en el diseño y presentación de la investigación para evitar largos procesos de revisión. También recomiendo que en estudios de este tipo ayudes a las explicaciones con tablas y figuras y detalles las explicaciones.